# Modeling of Proteolysis of β-Lactoglobulin and β-Casein by Trypsin with Consideration of Secondary Masking of Intermediate Polypeptides

**DOI:** 10.3390/ijms23158089

**Published:** 2022-07-22

**Authors:** Mikhail M. Vorob’ev

**Affiliations:** A. N. Nesmeyanov Institute of Organoelement Compounds, Russian Academy of Sciences, 28 ul. Vavilova, 119991 Moscow, Russia; mmvor@ineos.ac.ru

**Keywords:** protein degradation, proteolysis mechanisms, trypsin, tryptophan fluorescence, demasking kinetics

## Abstract

The opening of protein substrates during degradation by proteases and the corresponding exposure of their internal peptide bonds for a successful enzymatic attack, the so-called demasking effect, was studied for β-lactoglobulin (β-LG) and β-casein (β-CN) hydrolyzed by trypsin. Demasking was estimated by monitoring the redshift in intrinsic tryptophan fluorescence, characterizing the accessibility of polypeptide chains to aqueous medium. The secondary masking of intermediate polypeptides, giving an inverse effect to demasking, caused a restriction of the substrate opening. This led to the limitations in the red shift of fluorescence and the degree of hydrolysis with a long time of hydrolysis of β-LG and β-CN at a constant substrate concentration and reduced trypsin concentrations. The proposed proteolysis model included demasking of initially masked bonds in the protein globule or micelle, secondary masking of intermediate polypeptides, and their subsequent slow demasking. The hydrolysis of peptide bonds was modeled taking into account different hydrolysis rate constants for different peptide bonds. It was demonstrated that demasking competes with secondary masking, which is less noticeable at high trypsin concentrations. Modeling of proteolysis taking into account two demasking processes and secondary masking made it possible to simulate kinetic curves consistent with the experimental data.

## 1. Introduction

Enzymatic hydrolysis of proteins by proteolytic enzymes (proteolysis) leads to the degradation of the protein substrate and formation of a mixture of smaller polypeptides, short peptides and amino acids. Despite the importance of proteolysis for many areas of life science and biotechnology, the kinetic description of this complex phenomenon is still incomplete to allow, for example, quantitative prediction of the kinetics of the formation of individual peptides over time [1]. The difficulties arise even when describing proteolysis only by the most general characteristics—the degree of hydrolysis of peptide bonds and the total hydrolysis rate, if it is required to describe the process from the beginning to the end of proteolysis [2].

Proteolysis involves several processes, each of which occurs over time in accordance with its own laws. The hydrolysis of peptide bonds is the basis of proteolysis, leading to sequential cleavage of first specific and then less-specific peptide bonds. The inhibition of an enzyme by proteolysis products is also an important process determining the slowdown of proteolysis during the reaction due to reversible or irreversible binding of free enzymes. The destruction of the protein globule or micelle of the substrate begins the process of opening internal peptide bonds for a successful enzymatic attack [3,4]. This process, called demasking, which removes steric obstacles to the movement of the enzyme for the formation of a productive enzyme–substrate complex, is also an important part of proteolysis [5,6].

The specificity and mechanism of the hydrolysis by trypsin, which is a serine protease (EC 3.4.21.4), have been well-studied for the synthetic substrates with ester or amide bonds [7,8,9]. The optimum conditions for its action are pH 7.8 and temperature 37 °C [10]. Trypsin has a catalytic triad within its active site which involves Ser_195_, His_57_, and Asp_102_ amino acid residues. The primary specificity of trypsin is due to the interaction of the side chains of Arg or Lys residues of the substrate with Asp_189_ in the lower part of the active site of trypsin [11]. For small substrates with one hydrolysable bond, the hydrolysis kinetics obeys the Michaelis–Menten law.

In the protein substrates, trypsin predominantly cleaves peptide bonds at the carboxyl side of lysine and arginine (Lys-X and Arg-X bonds) unless they are followed by proline [9,10,11]. In addition, the rate of hydrolysis of these bonds depends also on other neighboring amino acid residues providing so-called secondary specificity. The total rate of hydrolysis of peptide bonds is expressed by the Michaelis–Menten equation only at the beginning of proteolysis [12]. An empirical exponential model determines the total rate of bond hydrolysis using only two parameters [13,14]. This model allows kinetic curves to be described over a sufficiently long range of proteolysis, where the exponential law is valid [15,16,17]. The kinetics of the hydrolysis of peptide bonds was studied systematically for a number of protein substrates and proteolytic enzymes at various concentrations of the substrate and enzyme. Hydrolysis of some peptide bonds during proteolysis is accompanied by the structural changes in the protein, which in turn predetermine the hydrolysis of other bonds. The changes in tertiary or/and secondary structure of protein substrate in the beginning of proteolysis and the subsequent conformational changes in polypeptide chains during proteolysis can be determined by spectral methods. Certain advances in quantitative registration of the conformational changes during proteolysis have been achieved using fluorescence [6,18] and infrared spectroscopy [19]. Thus, the structural changes in protein substrates were determined by the redshift in the maximum fluorescence of the tryptophan residues caused by an increase in the polarity of the medium around these residues during proteolysis [6,18].

Unlike hydrolysis of the low-molecular-weight substrates with one hydrolysable bond, proteolysis cleaves peptide bonds of different secondary specificity, while the accessibility of these bonds for the enzyme is not the same for different bonds and may vary in the result of their demasking during proteolysis [5]. A destruction of the original structure of the globular protein or protein aggregates (micelles) increases the accessibility of the remaining peptide bonds for the enzyme. This process provides a gradual demasking of peptide bonds [5,20], leading to an increase in the rate of hydrolysis when the initially masked bonds become demasked [5]. The opposite process was noted during proteolysis of β-CN by trypsin when increased aggregation and a local increase in masking were observed some time after the start of proteolysis [2]. The formation of the additionally masked peptide bonds from the proteolysis intermediates, as a result of their aggregation or conformational rearrangements, is referred to as the secondary masking.

The kinetics of bond hydrolysis, taking into account their demasking, was studied using the two-step proteolysis model [5,6,20]. In this model, proteolysis is considered as a two-step process with the sequential stages of demasking and hydrolysis. The rate of demasking was determined from the shift in the fluorescence of tryptophan residues, which change fluorescent properties as the protein globule degrades or protein micelles are destroyed [2,6]. The complication of the two-step proteolysis model was carried out for the proteolysis of β-LG with trypsin, taking into account two phases of demasking, corresponding to the degradation of the protein globule and the destruction of the remaining hydrophobic core [20].

The bovine whey protein, β-lactoglobulin (β-LG), the main protein in whey, transports fatty acids and vitamin A in vivo, and its preparations are widely used in the food industry due to their high nutritional value and good functional properties [21]. β-LG polypeptide chain contains 15 lysine residues and 3 arginine residues that are specific for trypsin, while its globule is stabilized by two disulfide bonds [22,23]. Proteolysis of β-LG by various proteases has been intensively studied, since some of the proteolysis products are physiologically active [24,25,26,27,28,29]. The same is true for total casein and its largest fraction, β-casein, β-CN [1,30,31]. β-CN is widely used for various physicochemical studies, since the micelles of β-CN are well-characterized and easy to use [32]. These studies made it possible to identify the main intermediate and final peptide fragments for proteolysis of β-LG and β-CN by trypsin. The peptide fragments of β-LG analyzed at various times of proteolysis were used to establish the sequence of fragmentation steps [24,33,34], and a system of differential equations was derived for describing these steps [33]. The relatively low concentrations of the protein substrates, such as 0.25 g/L in this work, guarantee the absence of the reverse reaction of proteolysis, the plastein reaction [35], leading to the formation of new peptide bonds.

The difficulties in the description of proteolysis are due to the complexity of proteolysis and the fact that some stages of this process may not be taken into account by existing models of proteolysis. An example of such stages can be non-enzymatic steps, the rate of which does not depend on the concentration of enzyme. The aggregation of the intermediate proteolysis products can isolate a part of peptide bonds from the enzyme, while the aggregation process itself occurs without the participation of enzyme. The peptides with unhydrolyzed specific peptide bonds were found in the peptide aggregates of intermediate proteolysis products of whey proteins, for example [36,37]. By changing the enzyme concentration, it is possible to change the rates of only enzymatic steps at the constant rates of the non-enzymatic steps and, thus, to regulate the proteolysis process. From this point of view, the data on the proteolysis of β-CN and β-LG by trypsin at various enzyme concentrations are analyzed in the present publication.

Here, we present a kinetic analysis of the data obtained from the hydrolysis of globular β-LG and structurally disordered β-CN with trypsin at various concentrations and a constant substrate concentration. Proteolysis was monitored by the measuring a shift of the spectra of tryptophan fluorescence, recording the dependence of the fluorescence maximum (λmax) from the hydrolysis time. During proteolysis, the degree of hydrolysis of peptide bonds was also recorded. In contrast to our previous work [18], the measurements were carried out at comparatively long times of proteolysis. The data processing consisted of determining the asymptotic values of λmax, when the time of proteolysis tends to infinity. The aim of the study was to analyze the dependence of the proteolysis parameters at the end of proteolysis on the concentration of trypsin. To explain these dependencies, a kinetic scheme was proposed, suggesting a competition between the demasking of peptide bonds as a result of the opening of protein substrates during proteolysis and the secondary masking of intermediate polypeptides, which gives an effect opposite to demasking.

## 2. Results

### 2.1. Monitoring of Proteolysis by Fluorescence Spectrometry

The fluorescence spectra were scanned during proteolysis with a time resolution of 1–2 min at the beginning of proteolysis and 5–15 min at the final stages of the process. The changes in the fluorescence spectra of β-LG and β-CN upon proteolysis with trypsin are shown in Figure 1.

During proteolysis, the position of the fluorescence maximum (λmax) shifts towards higher wavelengths from 340 nm to about 354 nm for β-LG and from 342 to about 358 nm for β-CN. This redshift was attributed to an increase in the polarity of the medium around Trp residues upon the protein opening during proteolysis [6]. In the beginning of proteolysis, the dependences λmax(t) for β-CN and β-LG are quite different [2]. For β-LG, λmax(t) monotonously increases, while for β-CN the dependence is decreasing within the first 5–10 min of the process [2]. The observed differences in the dependences λmax(t) at the beginning of proteolysis were explained by the difference between these protein substrates, which was discussed in detail earlier [2]. In the present publication, we analyze the dependences λmax(t) obtained at medium and relatively long proteolysis times up to 4 h (Figure 2).

At the end of the proteolysis process, when proteolysis time tends to infinity, the fluorescence maximum tends to an asymptote value *λ**. In the proteolysis experiments obtained during the prolonged proteolysis with both substrates, we observed that the values of *λ** increase with increasing trypsin concentration (Figure 2). This means that Trp residues in the peptide fragments that give different levels of fluorescence *λ** have different conformations, and the larger the value of *E*, the more these residues are exposed to the polar medium (water buffer).

In our previous study, the dependence of *λ*_max_ on the hydrolysis time was analyzed under the assumption of the presence of two demasking processes (Equation (10) in [18]). At long hydrolysis times, this equation can be transformed into a simpler equation with one exponential term (Equation (11)). The application of this equation to the final part of the functions *λ*_max_(*t*) (Figure 2) made it possible to determine *λ** and *k* for the proteolysis of β-LG and β-CN by trypsin (Table 1). Actually, the asymptote *λ**, indicating the value of *λ*_max_ at infinite time, is greater with the greater *E* value. This was found for both β-LG and β-CN substrates.

In the case of complete proteolysis, *λ** should be maximal, provided that all residues of Trp are fully exposed to the polar medium (water buffer). The highest values of *λ** were found for the high enzyme concentrations of 15 mg/L for β-LG and 2.5 mg/L for β-CN (Table 1). At lower enzyme concentrations (4.5 and 0.9 mg/L for β-LG, and 0.5 and 0.25 mg/L for β-CN), the fluorescence shift is smaller, which means that a part of Trp residues may be located more within the hardly hydrolysable nucleus or aggregates.

### 2.2. Proteolysis Monitoring Based on DH Determinations

Monitoring of the enzymatic hydrolysis of protein substrates is usually carried out by measuring the increase in the concentration of amine nitrogen during hydrolysis of peptide bonds and recalculating this value in the percentage of the hydrolyzed peptide bonds, the degree of hydrolysis (DH or *d* = DH/100) [32,33]. For both substrates, the changes in *d* versus hydrolysis time were determined for two enzyme concentrations (Figure 3). For both protein substrates, a general pattern was observed—the higher the enzyme concentration, the higher *d* values are obtained at the same times. With a long proteolysis time of up to 4 h, at which time the hydrolysis rate is significantly reduced, this pattern also remains (Figure 3).

Thus, both the data on the fluorescence of tryptophan residues and the data on the hydrolysis of peptide bonds show that an increase in the concentration of trypsin causes a deeper proteolysis.

### 2.3. Proteolysis Kinetic Scheme

Two processes were considered previously that make up demasking during proteolysis of β-LG by trypsin: one-stage demasking and two-stage demasking [18]. The first process was associated only with the unfolding of the protein globule and the second was associated with the additional demasking owing to destruction of a hard hydrolysable core of the remaining polypeptide chains of β-LG. A modification of the demasking mechanism, described earlier for the proteolysis of β-LG by trypsin [18], is proposed here with the consideration of the secondary masking.

The one-stage transition of the substrate from the masked to demasked state has been represented previously by the following scheme [18]: $S_{m}^{1}\overset{k_{1}E}{\to }S_{d}^{1}$, where $S_{m}^{1}$ stands for the masked state and $S_{d}^{1}$ stands for the demasked state, where the polypeptide chain is slightly open and the peptide bonds can be partially attacked by an enzyme, and $k_{1}E$ is the rate constant of this transition. After this transition, some of the polypeptide chains *S_mm_* can also be masked again owing to an aggregation, for example. Taking into account the secondary masking, the scheme of the one-stage demasking can be represented as:(1)$$S_{m}^{1}\overset{k_{1}E}{\to }S_{d}^{1}\overset{k_{m}}{\to }S_{mm}$$

The maximum fluorescence *λ*_max_ of the substrate in the masked and demasked states is *λ*_1_ and *λ*_2_, and it was assumed that the aggregated and non-aggregated states have the same values of *λ*_2_.

In the previous study, the two-stage transition of the substrate from the masked to demasked state was represented by the following scheme [18]: $S_{m}^{2}\overset{k_{1}E}{\to }S_{d}^{2}\overset{k_{2}E}{\to }S_{dd}^{2}$, where $S_{m}^{2}$ stands for the masked state, $S_{d}^{2}$ stands for the state where the polypeptide chain is slightly open but the peptide bonds still are unhydrolyzable, $S_{dd}^{2}$ stands for the completely open polypeptide chain with the completely hydrolyzable bonds, and $k_{1}^{}E$ and $k_{2}^{}E$ are the rate constants for the first and second stages of demasking. The maximum fluorescence *λ*_max_ of the substrate in a completely demasked state is *λ*_3_. The modified scheme with the presence of the secondary masking is:(2)$$S_{m}^{2}\overset{k_{1}E}{\to }\begin{array}{c} k_{m}^{} \end{array}\begin{array}{c} S_{d}^{2}\\ \begin{array}{c} ↓\\ S_{mm} \end{array} \end{array}\overset{k_{2}E}{\to }S_{dd}^{2}$$

In this scheme, $S_{mm}$ stands for the secondary masked polypeptide chains in which peptide bonds completely lose their ability to be hydrolyzed and $k_{m}^{}$ is the rate constant of the secondary masking.

The novelty lies in the fact that the transformation of $S_{dd}\overset{k_{m}}{\to }S_{mm}$ is not enzymatic and $k_{m}^{}$ does not depend on the concentration of enzyme. Conversely, the processes $S_{m}^{2}\overset{k_{1}E}{\to }S_{d}^{2}$ and $S_{d}^{2}\overset{k_{2}E}{\to }S_{dd}^{2}$ are enzymatic and the rate constants $k_{1}^{}E$ and
$k_{2}^{}E$
are assumed to be proportional to the concentration of the enzyme.

The hydrolysis of the peptide bonds, which are demasked in one stage, corresponds to the scheme:(3)$$B_{m}^{j}\overset{k_{1}E}{\to }\begin{array}{c} k_{m}^{} \end{array}\begin{array}{c} B_{d}^{j}\\ \begin{array}{c} ↓\\ B_{mm} \end{array} \end{array}\overset{k^{j}E}{\to }N_{1}^{j}$$

The hydrolysis of the peptide bonds, which are demasked in two stages, correspond to the following scheme:(4)$$B_{m}^{j}\overset{k_{1}E}{\to }\begin{array}{c} k_{m}^{} \end{array}\begin{array}{c} B_{d}^{j}\\ \begin{array}{c} ↓\\ B_{mm} \end{array} \end{array}\overset{k_{2}E}{\to }B_{dd}^{j}\overset{k^{j}E}{\to }N_{2}^{j}$$

For each peptide bond in the polypeptide chain with the number *j*, either Equation (3) or Equation (4) can be used depending on the mechanism of demasking of this bond, and *k^j^* is the rate constant of the hydrolysis of the *j*th bond.

### 2.4. Verification of Kinetic Scheme

The concentration dependences $S_{d}^{2}$ and $S_{dd}^{2}$ on *t* are given in Section 4.5 (Equations (8) and (9)). These functions are similar to those that were obtained earlier for the scheme without the secondary masking (Equations (7) and (8) in [18]) with the difference that the demasking rate constant for the second stage was $k_{2}E$, but it is now $k_{m}+k_{2}E$, and the asymptotic value for $S_{dd}^{2}(t)$ was $S_{0}^{2}$, but now it is $S_{0}^{2}k_{2}E/(k_{m}+k_{2}E)$ and depends on *E*. The same changes are made for the functions describing the concentration of the hydrolysis products (Equations (12) and (14)).

We carried out a numerical simulation of the dependences for a set of the parameters that were estimated taking into account the available experimental data. In particular, for the proteolysis of β-LG by trypsin, the ratio *k*_2_/*k*_1_ was taken equal to the ratio $k_{d}/k_{d}^{f}=0.3$ estimated earlier [18]. The linear dependence of *k* on the concentration of enzyme *k* = *k_m_* + *k*_2_*E* (Equation (11)) made it possible to estimate the *k_m_*/*k*_2_ ratio in the range of 0.2–0.3 mg/L. Thus, the range of the possible changes in the model parameters was narrowed.

In order to simplify the verification of the proposed model, the dimensionless time *τ* = *k*_1_*E*_0_*t*, the dimensionless enzyme concentration *ε* = *E*/*E*_0_, the dimensionless hydrolysis rate constants *κ^i^* = *k^i^*/*k*_1_, and the dimensionless rate constant of the secondary masking *κ_m_* = *k_m_*/(*k*_1_*E*_0_) were introduced. This made it possible to carry out calculations using equations similar to Equation (16), in which the parameters and the time variable were dimensionless quantities.

An example of *λ*_max_(*τ*) and *d*(*τ*) dependences is presented for two concentrations of the enzyme and two values of the parameter *κ_m_* (Figure 4). The dependences *λ*_max_(*τ*) were calculated using Equation (11) in the dimensionless form in the absence (*κ_m_* = 0) and in the presence of the secondary masking. For each of these cases, the calculations were performed at two enzyme concentrations (Figure 4a). Equation (16) was used to calculate *d*(*τ*) for a set of the kinetic parameters given in Section 4.5 (Figure 4b). In the absence of the secondary masking (*κ_m_* = 0), the curves *λ*_max_(*τ*) and *d*(*τ*) grow faster at a higher enzyme concentration. The asymptotes, representing the upper values of these functions when time tends to infinity, are the same at *κ_m_* = 0 regardless of the enzyme concentration. In the presence of the secondary masking, the curves *λ*_max_(*τ*) and *d*(*τ*) grow faster at the higher enzyme concentration, and the asymptotes for these functions increase with the increasing enzyme concentration.

In our calculations, for any value of *κ_m_* except *κ_m_* = 0, the values of *λ*_max_ and *d* at long hydrolysis times increased with increasing enzyme concentration. It is this pattern that we found in the experiments with proteolysis of β-LG and β-CN by trypsin (Figure 2 and Figure 3).

### 2.5. Simulation of Total Hydrolysis Kinetics

In the empirical exponential model of proteolysis, an exponential dependence of the rate of hydrolysis (*r* = d*d*/d*t*) on the degree of hydrolysis *r* = *a*·exp(−*b*·*d*) has been demonstrated for several protein substrates and enzymes [13,14,15]. This dependence can be represented in the logarithmic form ln(*r*) = ln(*a*) − *b*·*d*, which is a linear dependence with the slope *b* and intersection point ln(*a*). To find out if our data can fit this model, we presented our results in the coordinates: ln(*r*) versus *d*.

A differentiation of the dependence *d*(*t*) (Figure 4b) allowed the determination the dependence of the rate of hydrolysis *r*(*t*) on *t*. Then, time was excluded from both dependences ln(*r*(*t*)) and *d*(*t*), which made it possible to construct the dependence ln(*r*(*d*)) from *d*. The functions of the logarithm of the rate of hydrolysis on the degree of hydrolysis were built from experimental data and also simulated using Equations (12)–(15) (Figure 5). For the proteolysis of β-LG by trypsin, the experimental dependence ln(*r*(*d*)) in a certain range of *d* is almost linear, while at large *d* values, the curve drops sharply (Figure 5a). The calculated dependencies shown in Figure 5 behave in a similar way.

The processing of the curves consisted of determining the slope *b* of the linear function ln(*r*(*d*)) = −*b*·*d* + const in the fixed interval of *d*, taken in this study from 0 to 0.05. An estimate of the degree of hydrolysis at which a sharp drop in the logarithm of the hydrolysis rate is observed is illustrated in Figure 5b. This value, denoted by *d**, corresponds to a decrease in the initial rate of hydrolysis *r*_0_ by 10 times or when ln(*r*) becomes equal to ln(*r*_0_) − 2.303.

Figure 5b shows the dependences in the absence (*κ_m_* = 0) and presence (*κ_m_* = 0.025 and 0.05) of the secondary masking. Figure 5a shows the effect of the concentration of enzyme at constant *κ_m_* = 0.05. The linear part of the curves is longer, and the higher the enzyme concentration (Figure 5a), the lower the secondary masking rate (Figure 5b, Table 2).

For the experimental points of β-LG proteolysis by trypsin, the slope *b* was found to be equal to 22.5 ± 2.5 in the interval *d* from 0 to 0.05, and *d** was 0.067 (Figure 5a). For the theoretical dependencies, the close values of *b* from 18.6 to 28.7 and *d** from 0.081 to 0.057 were obtained at *κ_m_* = 0.05 for the enzyme concentrations in the interval from *ε* = 5 to 0.2 (Figure 5a). An increase in the coefficient *b* with a decrease in the concentration of enzyme was determined for the proteolysis of bovine serum albumin by trypsin and chymotrypsin using FTIR spectroscopy [19]. Due to the fact that the value of *b* changes relatively little with the changes in proteolysis conditions, in many studies that use the exponential proteolysis model, *b* is considered to be a constant value [14].

In order to understand how the correct value of the slope *b* for a linear part of the function ln(*r*(*d*)) can be achieved, the contribution to the total hydrolysis of the hydrolysis of various peptide bonds was considered. In Section 4.5, the equations for three types of peptide bonds were introduced, which describe the hydrolysis kinetics by Equations (12)–(14). The contributions to the degree of hydrolysis corresponding to these types are shown in Figure 6a, and the corresponding contributions to the hydrolysis rate are shown in Figure 6b. The first type corresponds to the rapid hydrolysis according to Equation (13) with the demasking rate constant *k*_1_, since the hydrolysis rate is not a limiting factor. The second one corresponds to the hydrolysis due to the one-stage demasking, leading to the appearance of peptide bonds open for the enzyme (Equation (12)). The third type of curves corresponds to the hydrolysis of peptide bonds after their two-stage demasking (Equation (14)). This type of curve is characterized by the presence of a noticeable lag phase [18].

The hydrolysis of masked bonds after their demasking leads to an increase in the rate of hydrolysis in a middle of the process. The contribution of these bonds to the total hydrolysis of peptide bonds partially compensates for the sharp decrease in the total rate of hydrolysis during hydrolysis, which could be for a set of the completely demasked bonds with various *κ**^i^* (*κ**^i^* = 1, 0.8, 0.4, 0.35, 0.2). This effect ensures that the slope of a linear part of the model curve (*b* = 18.6-28.7 for *κ**_m_* = 0.05) is close to that which is observed experimentally (*b* = 22.5). For hydrolysis of the completely demasked and identical bonds, the slope is less (*b* = 14.9).

## 3. Discussion

To date, several models of proteolysis have been proposed, each of which has its own assumptions to simplify this complex phenomenon. Among the models are the Linderstrom–Lang qualitative model of proteolysis [38], the empirical exponential model [13,14,15], the two-step proteolysis model [5,6], and other models. The model presented here is an improved two-step proteolysis model [5] with two demasking stages, as in [18], but also taking into account the secondary masking of intermediates that protect a part of the peptide bonds from enzymatic attack. After the destruction of the protein globule, the fragments of the polypeptide chain become mobile and interact within the chain or/and with other polypeptide chains. Over time, the peptide chains can take conformations that are not convenient for interaction with the active site of the enzyme or form aggregates, so that a part of peptide bonds can be masked. The formation of aggregates is one of the possible and most probable mechanisms of the secondary masking. If the cleavage of peptide bonds is accelerated by increasing the concentration of the enzyme, then the formation of such aggregates can be avoided, since the aggregates do not have enough time to form.

Typically, the proteolysis models take into account only those steps of proteolysis that occur with the participation of enzymes. The enzyme is involved in the degradation of the protein globule, causing hydrolysis of the primary site. Other peptide bonds are then hydrolyzed, and both of these processes are enzymatic. However, some peptide fragments are involved in the non-enzymatic secondary masking, and this process can be slow because the peptide chains undergo many conformational changes before they can form stable aggregates, and the rate of this process does not depend on the enzyme concentration. Thus, the enzyme does not participate in the secondary masking, although this process occurs with the proteolysis intermediate products. Since the rate of hydrolysis and the rate of secondary masking depend differently on the concentration of the enzyme, the ratio of the rates of these processes can be controlled by changing the enzyme concentration. Because of this, our model predicts a decrease in the depth of hydrolysis with decreasing enzyme concentration. This is the main difference between our model and other models of proteolysis.

A deceleration of proteolysis can be associated with both the equilibrium inhibition of the enzyme by proteolysis products and the relatively slow irreversible inactivation of the enzyme during proteolysis [39]. Only equilibrium inhibition of the enzyme does not explain why, at a sufficiently long time of proteolysis, some specific peptide bonds have not yet been hydrolyzed. The kinetics of the slow enzyme inactivation with a characteristic time comparable with the time of the proteolysis process itself was analyzed for the first time for the proteolysis of casein with chymotrypsin [39], and later to explain the deceleration of proteolysis in an exponential model of proteolysis [13]. In the model proposed here, the retardation of proteolysis was associated with the conformational rearrangements and aggregation of the intermediate peptides, the hydrolysis of which becomes difficult, since some peptide bonds become masked at the intermediate stages of the process. In fact, both the slow inactivation of the enzyme and the secondary masking of peptide bonds can be realized simultaneously, and additional research is needed to study this complex process. In this research, it is necessary to study the limitations in the course of proteolysis of the relative mobility of the active site of the enzyme and the hydrolysable sites of the polypeptide substrate relative to each other.

The problem of the identification and quantitative determination of the peptides in hydrolysates is currently being solved using modern HPLC-MS methods [40,41]. Using HPLC-MS data, the degree of hydrolysis of any specific peptide bond can be calculated by summing the concentrations of all peptides, which are originated from the cleavage of this bond. This method was used to determine the selectivity parameters for the hydrolysis of peptide bonds [40,41]. This makes it possible to move from the description of proteolysis as a process of the continuous formation of many peptide fragments to the description of proteolysis as a process of the hydrolysis of individual bonds. It was also assumed that the effect of demasking can be determined from the characteristic time dependences of the concentrations of individual peptide bonds on the time of hydrolysis, since the cleavage of different bonds is interdependent during demasking [42]. The proteolysis of the serum proteins by *Bacillus licheniformis* protease was analyzed in terms of hydrolysis of individual bonds [42], and it was shown that more than a half of the kinetic curves have a characteristic shape, indicating the presence of demasking effect [42].

We have shown that by changing *E*, one can not only speed up or slow down the entire process, but also change the depth of hydrolysis. A quantitative prediction of this was made using the equations derived here, which contain a fairly significant number of parameters that are valid for a given protein substrate and enzyme. From the mathematical considerations, we assert that the qualitative regularities found here are valid for any values of the parameters, i.e., they must be valid for any enzyme–substrate pair. Therefore, it can be expected that the higher the concentration of the enzyme, the deeper the hydrolysis will be for any enzyme–substrate pair. However, in order to know exactly how much deeper, it is necessary to know the values of the parameters specifically for the given case of proteolysis. For the proteolysis of β-LG by trypsin, the estimation of parameters *k_m_* and *k*_2_/*k*_1_ was presented above. It is not so easy to estimate the relative rate constants of hydrolysis *κ^i^* for the different peptide bonds. To evaluate 15 specific peptide bonds out of 18 Lys-X and Arg-X bonds in β-LG, the selectivity parameters from [41] were used, and the hydrolysis rate constants of the remaining three hardly hydrolysable bonds (K47, K60, and K100) were taken equal to zero. Since the selectivity parameters are the apparent hydrolysis rate constants, they could be used only for a ranking purpose. The peptide bonds K8, K69, K75, and R148 were assigned to the group of bonds with the one-stage demasking and *κ^i^* >> 1, the peptide bonds R40 and K141 to the group with *κ^i^* = 0.8, the peptide bonds K14 and K70 to the group with *κ^i^* = 0.4, and the peptide bonds R124 and K138 to the group with *κ^i^* = 0.2. The peptide bonds K77, K83, K91, K101 and K135 were assigned to the group in which hydrolysis proceeds after the two-stage demasking with *κ^i^* = 0.35. Knowing the exact values of the hydrolysis rate constants for various peptide bonds is necessary for a successful modeling of proteolysis, and these studies need to be continued.

The modeling of proteolysis is usually carried out using the solutions of small proteins such as β-LG and β-CN, since the limited number of specific bonds in these substrates suggests a moderate number of model parameters [1,27,30,38]. This traditional approach was used in our study for a rather dilute protein concentration (0.25 g/L). For large proteins, which may have a quaternary structure or even more complex spatial organization, especially at high concentrations, the approach developed here cannot be used directly due to the extreme complexity of such systems and a need for an additional consideration of the processes not taken into account here. A possibility of complicating the model can be considered if a sufficient amount of quantitative data on the proteolysis of large proteins and protein complexes is accumulated. 

The proteolytic enzymes in vivo perform both a destructive function, carrying out the breakdown of proteins, and a regulatory function, controlling cellular metabolism. The regulatory role is associated with the activation of the enzymes and biologically active peptides by hydrolysis of only one peptide bond, as a rule. This limited proteolysis is of biological importance because it releases the biologically active fragment of the polypeptide chain but keeps them unhydrolyzed. The strict restriction of the hydrolysis of many peptide bonds is usually explained by the fact that the conformation of the protein substrate allows the enzyme to access only a certain target site. In the herein presented example of in vitro proteolysis of β-LG and β-CN by trypsin, a much larger number of peptide bonds are hydrolyzed; however, as the enzyme concentration decreases, the number of hydrolyzable bonds reduces. Certainly, we did not observe a decrease in the number of hydrolyzable bonds to a few within the range of the trypsin concentrations used, but the effect of narrowing the specificity of trypsin with decreasing its concentration was definitely observed. Moreover, this effect, apparently, is not so sensitive to the structure of the polypeptide substrate and is characteristic of proteolysis itself due to the universality of Equations (1)–(4). Taking into account the mechanisms of the competition between demasking and secondary masking could concretize the concept of limited proteolysis. Further research is needed to see how useful the model of proteolysis proposed here can be for the various protein systems of biological importance.

## 4. Materials and Methods

### 4.1. Materials

β-LG (L3908), β-CN (C6905) from bovine milk, and trypsin from bovine pancreas (T1426) treated with N-tosyl-L-phenylalanine chloromethyl ketone (TPCK) were purchased from Sigma-Aldrich and used without further treatment. TPCK was used to inhibit the contaminating chymotrypsin activity without affecting the activity of trypsin. The phosphate-buffered solution was prepared with doubly distilled water and stored at 4 °C before use. Trypsin solutions in phosphate buffer were freshly prepared by diluting the freeze-dried trypsin with activity of 9.8 BAEE (N-benzoyl-L-arginine ethyl ester) units per µg of trypsin. All other reagents were of analytical grade and obtained from commercial sources.

### 4.2. Proteolysis Reaction

The protein substrate (β-LG or β-CN) was dissolved in 20 mM phosphate buffer (pH 7.9) at 37 °C by stirring. The enzymatic hydrolysis was carried out in a bath-stirred reactor for the determination of amino nitrogen or in a 1 cm quartz cuvette for the fluorescence measurements at a constant concentration of the substrate and various concentrations of the enzyme. For example, enzymatic hydrolysis in a reactor volume of 2 mL with a substrate concentration of 0.25 g/L was initiated by adding 10 µL of stock trypsin solution (1 g/L) to provide the trypsin concentration in the reaction mixture of 5.0 mg/L.

### 4.3. Fluorescence Spectroscopy and Determination of Demasking Kinetics by Fluorescence Measurements

Fluorescence emission during proteolysis was measured using a Perkin-Elmer LS 55 Luminescence Spectrometer (Waltham, MA, USA) at 90° relative to the excitation beam at an excitation wavelength of 280 nm. The spectral bandwidth of the excitation and emission light was set to 10 and 5 nm, respectively. A thermostated cuvette holder with a magnetic stirrer was used to keep the sample at 37 °C. The emission spectra were recorded at a scanning speed of 150 nm/min.

To determine the wavelength of the fluorescence maximum, we used a parabolic function *I*(*λ*) = *a**λ*^2^ + *b**λ* + const for the approximation of the fluorescence spectrum, taking only a small area with a bandwidth of 30 nm. The parabolic function was used to approximate the fluorescence spectrum in a small region of 30 nm around peak maximum, which allowed us the determination of *λ*_max_—the position of the parabola center at *λ*_max_ = −*b*/2*a* [6].

### 4.4. Determination of Hydrolysis Degree by OPA Method

The OPA method of amino nitrogen determination is based on the reaction of *o*-phthalaldehyde (OPA) and 2-mercaptoethanol with amino groups released during proteolysis [43]. A stock OPA solution was prepared by mixing 25 mL of 100 mM sodium tetra hydroborate, 2.5 mL of 20% SDS solution (*w*/*w*), 40 mg of OPA and 100 µL of β-mercaptoethanol. The final volume of the OPA solution (50 mL) was adjusted with distilled water.

For the termination of proteolysis after a required time *t* and for monitoring of the hydrolysis reaction by the OPA-method, a 20 µL sample from the proteolysis reactor was placed in a test tube with 1 mL of OPA solution, incubated for 5 min at room temperature, and then the absorption at 340 nm was determined against a blank sample. The latter was prepared in the same way except for the absence of the enzyme and substrate. In the results, the value of amino nitrogen *N*(*t*) was determined at time *t*. For the determination of amino nitrogen at the hydrolysis of all peptide bonds (Δ*N*), a complete hydrolysis of the protein substrates in 6N HCl at 110 °C under argon during 24 h was performed. The degree of hydrolysis at time *t* was calculated as *d* = (*N*(*t*) − *N*(0))/Δ*N*.

### 4.5. Quantitative Modelling of Proteolysis

The concentration dependences of $S_{m}^{1}$ and $S_{d}^{1}$ on *t* are similar to our previous study [18]:(5)$$S_{m}^{1}=S_{0}^{1}e^{-k_{1}Et}$$
(6)$$S_{d}^{1}=S_{0}^{1}(1-e^{-k_{1}Et})$$

Equation (2) corresponds to the following concentration functions of *t*:(7)$$S_{m}^{2}=S_{0}^{2}e^{-k_{1}Et}$$
(8)$$S_{d}^{2}=\frac{S_{0}^{2}k_{1}E}{k_{1}E-k_{m}-k_{2}E}\left[-e^{-k_{1}Et}+e^{-(k_{m}+k_{2}E)t}\right]$$
(9)$$S_{dd}^{2}=\frac{S_{0}^{2}k_{2}E}{k_{m}+k_{2}E}\left[1-\frac{k_{m}+k_{2}E}{k_{m}+k_{2}E-k_{1}E}e^{-k_{1}Et}+\frac{k_{1}E}{k_{m}+k_{2}E-k_{1}E}e^{-\left(k_{m}+k_{2}E\right)t}\right]$$

The theoretical dependence *λ*_max_ on *t* can be calculated as:(10)$$\lambda_{\max}=(S_{m}^{1}\lambda_{1}+S_{d}^{1}\lambda_{2}+S_{m}^{2}\lambda_{1}+S_{d}^{2}\lambda_{2}+S_{dd}^{2}\lambda_{3})/S_{0}$$
where *λ*_1_, *λ*_2_ and *λ*_3_ were estimated for the proteolysis of β-LG by trypsin [18].

The concentrations $S_{m}^{1}$, $S_{d}^{1}$, $S_{m}^{2}$, $S_{d}^{2}$ and $S_{dd}^{2}$ from Equations (5)–(9) were introduced into Equation (10), and the obtained equation was simplified for the long times (*t* >> 1/*k*_1_*E*):(11)$$\lambda_{\max}(t)\approx \frac{S_{0}^{1}}{S_{0}}\lambda_{2}+\frac{S_{0}^{2}}{S_{0}}\frac{k_{2}E\lambda_{3}}{k_{m}+k_{2}E}-\frac{S_{0}^{2}}{S_{0}}\frac{k_{1}E}{k_{1}E-k_{m}-k_{2}E}(\lambda_{3}\frac{k_{2}E}{k_{2}E+k_{m}}-\lambda_{2})e^{-(k_{m}+k_{2}E)t}=\lambda^{*}-ae^{-(k_{m}+k_{2}E)t}$$
where *λ**, *a* and $k=k_{m}+k_{2}E$ are the parameters that can be determined experimentally.

With the one-stage demasking, the concentration of the products resulting from the cleavage of the *i*th bond can be calculated as follows:(12)$$N_{1}^{i}(t)=N_{0}^{i}\frac{k^{i}E}{k_{m}+k^{i}E}\left[1-\frac{(k_{m}+k^{i}E)e^{-k_{1}Et}}{k_{m}+k^{i}E-k_{1}E}+\frac{k_{1}Ee^{-(k_{m}+k^{i}E)t}}{k_{m}+k^{i}E-k_{1}E}\right]$$

If the hydrolysis rate constant is much greater than the demasking rate constant (*k^i^* >> *k*_1_), then Equation (12) is greatly simplified:(13)$$N_{1}^{i}(t)=N_{0}^{i}(1-e^{-k_{1}Et})$$

With the two-stage demasking, the concentration of the cleavage products of the *j*th bond can be calculated as follows:(14)$$N_{2}^{j}(t)=N_{0}^{j}\frac{k_{2}E}{k_{m}+k_{2}E}\left[1-\frac{(k_{m}+k_{2}E)k^{j}Ee^{-k_{1}Et}}{\left(k_{m}+k_{2}E-k_{1}E)(k^{j}E-k_{1}E\right)}-\frac{k_{1}Ek^{j}Ee^{-(k_{m}+k_{2}E)t}}{\left(k^{j}E-k_{m}-k_{2}E)(k_{1}E-k_{m}-k_{2}E\right)}-\frac{(k_{m}+k_{2}E)k_{1}Ee^{-k^{j}Et}}{\left(k^{j}E-k_{1}E)(k^{j}E-k_{m}-k_{2}E\right)}\right]$$

The change in the degree of hydrolysis of peptide bonds with hydrolysis time *t* can be calculated using Equations (12)–(14):(15)$$d(t)=\left(\sum_{i}N_{1}^{i}(t)+\sum_{j}N_{2}^{j}(t)\right)/B_{0}$$
where *B*_0_ is the initial concentration of all peptide bonds in the substrate. The summation over index *i* is carried out for the peptide bonds that are demasked in one stage, and the summation over the *j* index is carried out for the bonds that are demasked in two stages.

When modeling proteolysis, we introduced the dimensionless time *τ* = *k*_1_*E*_0_*t*, the dimensionless enzyme concentration *ε* = *E*/*E*_0_, and the dimensionless rate constant of the secondary masking *κ_m_* = *k_m_*/(*k*_1_*E*_0_). Assuming that *E*_0_, the average value of the enzyme concentration used in the work is equal to 10 mg/L, and the parameter *κ_m_* is 0.25 × 3.33/10 = 0.083 with the experimental values of *k_m_*/*k*_2_ = 0.25 mg/L and *k*_2_/*k*_1_ = 0.3. It was assumed that there are 161 peptide bonds in the substrate, of which 15 bonds are specific for the enzyme. The dimensionless constants of hydrolysis rate for the various peptide bonds *i* are equal to *κ^i^* = *k^i^*/*k*_1_. The distribution of the specific peptide bonds in the protein substrate was accepted to be as follows: 4 bonds with *k^i^* >> *k*_1_ are hydrolyzed by a simple exponential law with the relative rate constant equal to 1; 6 bonds are demasked in one stage and hydrolyzed with *k^i^*/*k*_1_ = 0.8, 0.4 or 0.2; and 5 bonds are demasked in two stages (*k*_2_/*k*_1_ = 0.3) and hydrolyzed with *k^i^*/*k*_1_ = 0.35. Thus, the hydrolysis of 15 specific bonds was depicted by the following equations: 4 bonds were hydrolyzed according to Equation (13); 6 bonds were hydrolyzed according to Equation (12); and 5 bonds were hydrolyzed according to Equation (14). For the arbitrary secondary masking rate constant *κ_m_* and the dimensionless enzyme concentration *ε*, the degree of hydrolysis of peptide bonds at the dimensionless time *τ* is:(16)$$\begin{array}{l} d(t)=\{4(1-e^{-1\varepsilon \tau })+2\frac{0.8\varepsilon }{\kappa_{m}+0.8\varepsilon }\left[1-\frac{(\kappa_{m}+0.8\varepsilon )e^{-1\varepsilon \tau }}{\kappa_{m}+0.8\varepsilon -1\varepsilon }+\frac{1\varepsilon e^{-(\kappa_{m}+0.8\varepsilon )\tau }}{\kappa_{m}+0.8\varepsilon -1\varepsilon }\right]\\ +2\frac{0.4\varepsilon }{\kappa_{m}+0.4\varepsilon }\left[1-\frac{(\kappa_{m}+0.4\varepsilon )e^{-1\varepsilon \tau }}{\kappa_{m}+0.4\varepsilon -1\varepsilon }+\frac{1\varepsilon e^{-(\kappa_{m}+0.4\varepsilon )\tau }}{\kappa_{m}+0.4\varepsilon -1\varepsilon }\right]+2\frac{0.2\varepsilon }{\kappa_{m}+0.2\varepsilon }\left[1-\frac{(\kappa_{m}+0.2\varepsilon )e^{-1\varepsilon \tau }}{\kappa_{m}+0.2\varepsilon -1\varepsilon }+\frac{1\varepsilon e^{-(\kappa_{m}+0.2\varepsilon )\tau }}{\kappa_{m}+0.2\varepsilon -1\varepsilon }\right]\\ +5\frac{0.3\varepsilon }{\kappa_{m}+0.3\varepsilon }\left[1-\frac{(\kappa_{m}+0.3\varepsilon )0.35\varepsilon e^{-1\varepsilon \tau }}{\left(\kappa_{m}+0.3\varepsilon -1\varepsilon )(0.35\varepsilon -1\varepsilon \right)}-\frac{1\varepsilon 0.35\varepsilon e^{-(\kappa_{m}+0.3\varepsilon )\tau }}{\left(0.35\varepsilon -\kappa_{m}-0.3\varepsilon )(1\varepsilon -\kappa_{m}-0.3\varepsilon \right)}-\frac{(\kappa_{m}+0.3\varepsilon )1\varepsilon e^{-0.35\varepsilon \tau }}{\left(0.35\varepsilon -1\varepsilon )(0.35\varepsilon -\kappa_{m}-0.3\varepsilon \right)}\right]\}/161 \end{array}$$

Similarly, Equation (11) for λmax can also be converted to the dimensionless form.

## Figures and Tables

**Figure 1 ijms-23-08089-f001:**
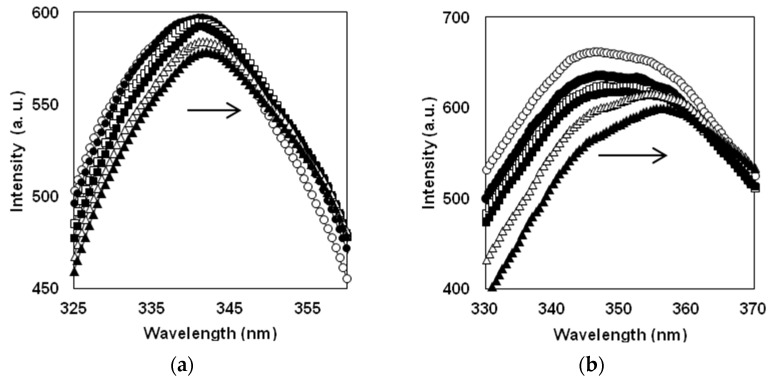
Change in fluorescent spectra in course of proteolysis: (**a**) proteolysis of β-LG (0.25 g/L) by trypsin (0.9 mg/L) at 0 (○), 10 (●), 20 (□), 30 (■), 40 (∆), and 50 min (▲); (**b**) proteolysis of β-CN (0.25 g/L) by trypsin (0.25 mg/L) at 0 (○), 3 (●), 5 (□), 7.5 (■), 23 (∆), and 45 min (▲). Arrows show the direction of the fluorescence shift during proteolysis.

**Figure 2 ijms-23-08089-f002:**
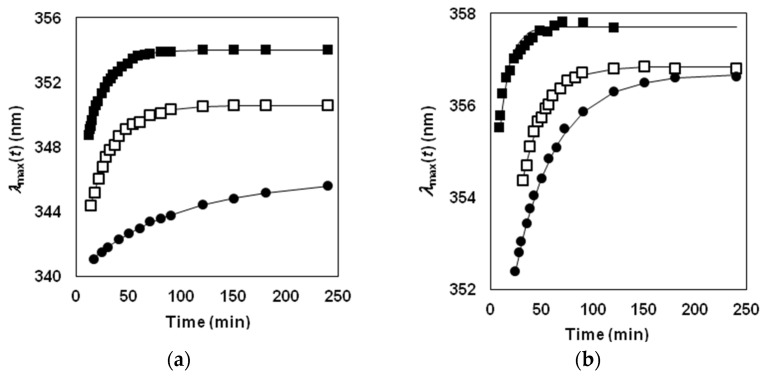
Dependences of the wavelength of the maximum fluorescence on proteolysis time: (**a**) proteolysis of β-LG (0.25 g/L) by trypsin at a concentration of 15 (■), 4.5 (□) and 0.9 mg/L (●); (**b**) proteolysis of β-CN (0.25 g/L) by trypsin at a concentration of 2.5 (■), 0.5 (□) and 0.25 mg/L (●). Solid lines correspond to the fitting of experimental points using Equation (11).

**Figure 3 ijms-23-08089-f003:**
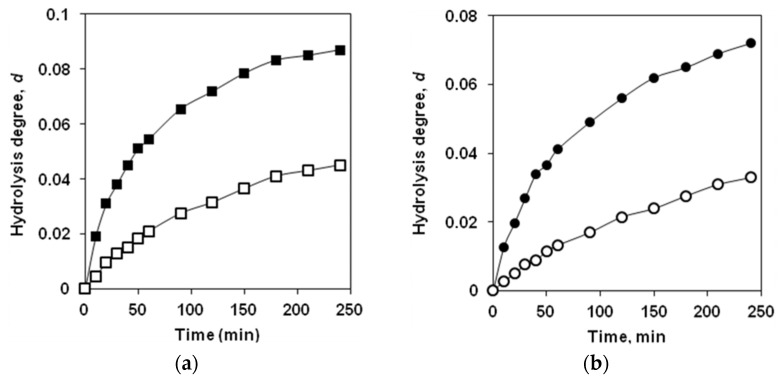
Dependences of the degree of hydrolysis on proteolysis time: (**a**) proteolysis of β-LG (0.25 g/L) by trypsin at a concentration of 5 (■) and 1 mg/L (□); (**b**) proteolysis of β-CN (0.25 g/L) by trypsin at a concentration of 0.5 (●), and 0.1 mg/L (○).

**Figure 4 ijms-23-08089-f004:**
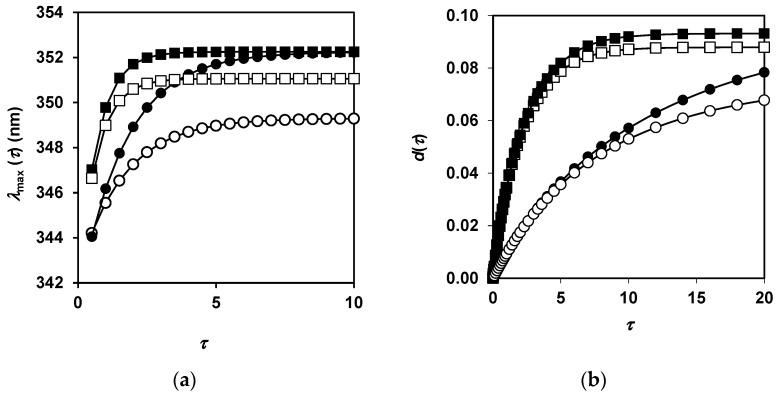
Numerical simulation of proteolysis process according to proposed model: (**a**) expected dependences of the fluorescence maximum on the dimensionless time *τ*at *ε* = 5 for *κ_m_* = 0 (■) and 0.01 (□), and at *ε* = 2.1 for *κ_m_* = 0 (●) and 0.01 (○); (**b**) expected dependences of the hydrolysis degree on the dimensionless time at *ε* = 2 for *κ_m_* = 0 (■) and 0.05 (□), and at *ε* = 0.4 for *κ_m_* = 0 (●) and 0.05 (○).

**Figure 5 ijms-23-08089-f005:**
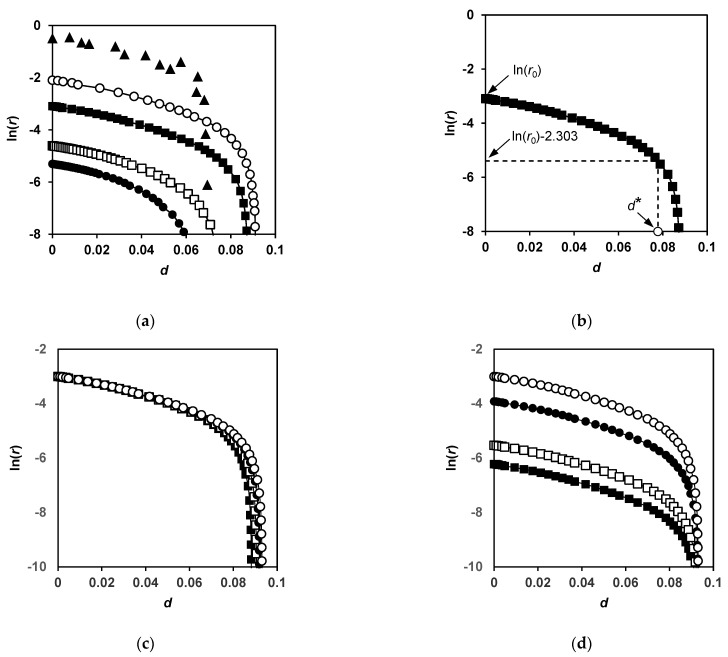
Dependences of the logarithm of hydrolysis rate ln(*r*) on degree of hydrolysis *d*: (**a**) experimental data (▲) and calculated dependences for *κ_m_* = 0.05 at *ε* = 0.2 (●), 0.4 (□), 2 (■), and 5 (○); (**b**) illustration of the definition of *d** value (○); (**c**) simulated dependences for *ε* = 2 at *κ_m_* = 0.05 (■), 0.025 (□), 0.012 (●), and 0 (○); (**d**) simulated dependences for *κ_m_* = 0 at *ε* = 0.2 (■), 0.4 (□), 2 (●), and 5 (○).

**Figure 6 ijms-23-08089-f006:**
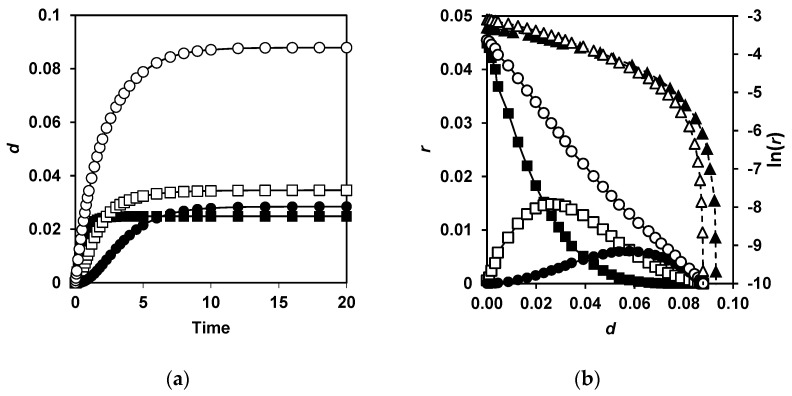
Contributions to the total hydrolysis (○) of three types of peptide bonds: 4 bonds (■) with one-stage demasking at *κ^i^* >> 1; 6 bonds (□) with one-stage demasking at *κ^i^* = 0.8 (2 bonds), *κ^i^* = 0.4 (2 bonds), and *κ^i^* = 0.2 (2 bonds); 5 bonds (●) with two-stage demasking at *κ^i^* = 0.35 and *k*_2_/*k*_1_ = 0.3. (**a**) Dependences of the degree of hydrolysis on hydrolysis time. (**b**) Dependences of the hydrolysis rate and the logarithm of hydrolysis rate on degree of hydrolysis. Total hydrolysis rate (○) and the logarithm of total hydrolysis rate (∆) for proposed model and the logarithm of total hydrolysis rate for completely demasked and identical bonds (▲).

**Table 1 ijms-23-08089-t001:** Fitting of fluorescence data with Equation (11).

Substrate	Concentration of Trypsin (mg/L)	*λ** (nm)	*k* (min^−1^)	*r* ^2^
β-LG	15	353.7 ± 0.3	0.065 ± 0.003	0.997
β-LG	4.5	352.0 ± 0.2	0.019 ± 0.002	0.998
β-LG	0.9	344.6 ± 0.2	0.006 ± 0.001	0.998
β-CN	2.5	357.8 ± 0.3	0.15 ± 0.05	0.994
β-CN	0.5	356.8 ± 0.2	0.040 ± 0.04	0.996
β-CN	0.25	356.4 ± 0.2	0.023 ± 0.03	0.997

**Table 2 ijms-23-08089-t002:** Simulation parameters for the modeling of total hydrolysis.

Secondary Masking Rate Constant (*κ_m_*)	Concentration of Trypsin (*ε*)	*b* ^1^	*d**
0.05	5	18.6 ± 0.7	0.081
	2	19.6 ± 0.6	0.077
	0.4	23.1 ± 0.7	0.066
	0.2	28.7 ± 1.1	0.057
0.025	5	18.4 ± 0.7	0.081
	2	19.5 ± 0.6	0.080
	0.4	20.3 ± 0.5	0.073
	0.2	22.9 ± 0.5	0.066
0	5; 2; 0.4; 0.2	18.4 ± 0.5	0.082

^1^ Linearization of the dependence ln(*r*) on *d* for the determination of *b* was carried out in the interval *d* from 0 to 0.05.

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
