# Peer review of "Modeling of Proteolysis of β-Lactoglobulin and β-Casein by Trypsin with Consideration of Secondary Masking of Intermediate Polypeptides"

_ijms, 2022, doi:10.3390/ijms23158089_

Round 1

Reviewer 1 Report

Dear Author,

The paper you proposed presents an interesting new model to study and predict the proteolysis process. More parameters were considered here that were not explored in previous literature. 

I consider this draft deserving of publication. 

Best wishes

Author Response

I would like to thank the reviewer for a very careful consideration of the manuscript.

The grammatical correction of the text was produced.

Reviewer 2 Report

This manuscript by Vorob’ev et. al. demonstrated that the demasking competes with secondary masking, which is less noticeable at high trypsin concentrations. Modeling of proteolysis taking into account two demasking processes and secondary masking made it possible to simulate kinetic curves consistent with the experimental data. The whole manuscript was well-organized, and the information provided in this study and the experimental methodology is interesting. Hence, I recommend its publication in IJMS after a minor revision with the following comments addressed.

minor point

The conclusion looks fine, and the main limitation also should be discussed as well.

There are some grammatical errors in this manuscript such as continuously forgetting to add ‘a’ or ‘the’ before a specific word which limits the clarity of the author’s writing. Check the language issues.

Author Response

I would like to thank the reviewer for a very careful consideration of the manuscript.

The following paragraph was added to the text to show the limitation of the model:

The modeling of proteolysis is usually carried out using the solutions of small proteins such as b-LG and b-CN, since the limited number of specific bonds in these substrates suggests a moderate number of the model parameters [1,27,30,38]. This traditional approach was used in our study for a rather dilute protein concentration (0.25 g/l). For the large proteins, which may have a quaternary structure or even more complex spatial organization, especially at high concentrations, the approach developed here cannot be used directly due to the extreme complexity of such systems and a need for an additional consideration of the processes not taken into account here. A possibility of the complicating the model can be considered if a sufficient amount of quantitative data on the proteolysis of large proteins and protein complexes is accumulated.

The grammatical correction of the text was produced.

Reviewer 3 Report

1.Could the author provide some photographs to express the proteolysis of b-lactoglobulin and b-casein?

2.Why the author want to discuss the  quantitative modelling of proteolysis, because it was original known equation?

Author Response

I would like to thank the reviewer for a very careful consideration of the manuscript.

1. Proteolysis is carried out in a conventional thermostated reactor using a conventional thermostat and a magnetic stirrer. I believe there is no need to provide a picture as this is not a unique setup.

The paper already presents graphs of the changes in the fluorescence maximum and the degree of hydrolysis of peptide bonds during proteolysis for both substrates (Figures 2 and 3). It is these graphs that reflect the essence of the processes occurring during proteolysis.

2. The mathematical equations obtained in the article (Equations 5-16) are unique because they reflect the kinetic scheme (Equations 1-4), which is proposed for the first time. A part of our research is that our equations are used to validate a well-known empirical equation describing proteolysis in the terms of the exponential model.

Reviewer 4 Report

In this paper, the author proposes a new kinetic model for proteolysis and tests it with experimental data for two proteins, \beta-LG, and \beta-CN. In contrast to previously existing models, the novelty of this work is that it includes two desmaking stages, being the second one due to the formation of aggregates of the hydrolyzed peptide chains, which is enzyme independent and slows down the overall proteolysis rate.

I find the work well designed, the paper well written, and the model well elaborated and sufficiently contrasted against the experimental data. The only minor critique is that it is perhaps a bit "niche" in the sense that it focuses on a very particular problem, and shows the validity of the model on two specific protein systems. The manuscript would perhaps benefit from a broader discussion, not only on the implications of the model for the studied protein systems but also discussing its validity (and limitations) for general protein systems, including in a biological context. Regardless of that, I believe that the paper is suitable for publication.

Author Response

I would like to thank the reviewer for a very careful consideration of the manuscript.

The following paragraph was added to the discussion chapter: 

The proteolytic enzymes in vivo perform both a destructive function, carrying out the breakdown of proteins, and a regulatory function, controlling cellular metabolism. The regulatory role is associated with the activation of the enzymes and biologically active peptides by hydrolysis of only one peptide bond, as a rule. This limited proteolysis is of biological importance because it releases the biologically active fragment of the polypeptide chain, but keeps them unhydrolyzed. The strict restriction of the hydrolysis of many peptide bonds is usually explained by the fact that the conformation of the protein substrate allows the enzyme to access only a certain target site. In the presented here example of in vitro proteolysis of b-LG and b-CN by trypsin, a much larger number of peptide bonds are hydrolyzed, however, as the enzyme concentration decreases, the number of hydrolyzable bonds reduces. Certainly, we did not observe a decrease in the number of hydrolysable bonds to a few within the range of the trypsin concentrations used, but the effect of narrowing the specificity of trypsin with decreasing its concentration was definitely observed. Moreover, this effect, apparently, is not so sensitive to the structure of the polypeptide substrate and is characteristic of proteolysis itself due to the universality of Equations (1-4). Taking into account the mechanisms of the competition between demasking and secondary masking could concretize the concept of limited proteolysis. Further research is needed to see how useful the model of proteolysis proposed here can be for the various protein systems of biological importance.

Round 2

Reviewer 3 Report

The author had corrected the manuscript according to my comments.

Author Response

Thank you.